# Self-Assembled Nanocarriers Based on Modified Chitosan for Biomedical Applications: Preparation and Characterization

**DOI:** 10.3390/polym12112593

**Published:** 2020-11-04

**Authors:** Alina Gabriela Rusu, Aurica P. Chiriac, Loredana Elena Nita, Irina Rosca, Daniela Rusu, Iordana Neamtu

**Affiliations:** 1Laboratory of Inorganic Polymers, “Petru Poni” Institute of Macromolecular Chemistry, 41-A Grigore Ghica Voda Alley, 700487 Iasi, Romania; rusu.alina@icmpp.ro (A.G.R.); lnazare@icmpp.ro (L.E.N.); rusu.daniela@icmpp.ro (D.R.); neamtui@icmpp.ro (I.N.); 2Center of Advanced Research in Bionanoconjugates and Biopolymers, “Petru Poni” Institute of Macromolecular Chemistry, 41-A Grigore Ghica Voda Alley, 700487 Iasi, Romania; rosca.irina@icmpp.ro

**Keywords:** chitosan derivatives, bovine serum albumin, self-assembly, nanogels

## Abstract

Protein-polysaccharide systems are of increasing interest as their combined attributes allow for fulfilling a broad range of applications in biomedical and pharmaceutical fields. In this study, the preparation of nanogels based on maleic anhydride chitosan derivatives (MAC) and bovine serum albumin (BSA) was achieved through a self-assembly process performed in aqueous phase. A series of experiments performed by varying the concentrations of MAC and BSA were conducted to find an appropriate mixing ratio for the polymer solutions leading to thermodynamically stable nanogels with the ability to encapsulate active compounds. The influence of temperature on the formation of nanogels was also studied. The consequent conformational changes were monitored using ultraviolet-visible (UV-VIS) spectrophotometry. The spectrophotometric investigations combined with diffraction light scattering (DLS) technique and zeta potential measurement results were correlated to determine the interaction mechanism and assess the self-assembling processes during nanogel formation. It was found that the hydrodynamic diameter (D_h_) of the nanoparticles increased slightly at acidic pH, and the protonation of ionizable amino groups with the pH was confirmed by the zeta potential measurements. MAC/BSA nanogels also exhibited antimicrobial properties after being loaded with amoxicillin (Amox), which is an antibiotic used for the treatment of various infections. The experimental data resulting from this study provide theoretical guidance for the design and development of attractive nanocarriers for a large variety of biomedical applications.

## 1. Introduction

Nanogels, an important class of biomaterials, have attracted interest in drug release due to their unique properties, such as small size and large surface area, which help them cross biological barriers (including the cell membrane) and accumulate in target sites, increasing the efficiency of the treatment with active principles [1]. Among the types of nanogels, those based on polysaccharides are of great interest in medical applications due to their biological properties such as biodegradability, biocompatibility, mucoadhesion, low cost, and low toxicity [2]. Moreover, proteins or polypeptides, natural polyelectrolytes with unique functional properties, such as the ability to form gels and emulsions, are preferred for encapsulating bioactive compounds. Nanogels formed by self-assembly of natural polymers like proteins/polypeptides and polysaccharides have been shown to be suitable systems for encapsulation and controlled release of bioactive compounds [3].

The study on interactions between proteins and polysaccharides finds application in many engineering, biotechnological, and biomedical areas [4,5]. In biological systems, proteins and polysaccharides are involved in the organization of the living cells, and thus the interactions between these biopolymers through association can lead to the formation of a complex network in which cells can reside [6,7]. In contrast, their incompatibility may lead to cellular damages.

Depending on the biopolymer’s characteristics (i.e., reactive sites, protein/polysaccharide-type, the net protein charge, chain flexibility, molecular weight, and charge density), concentration and ratio, and synthesis conditions (i.e., pH, ionic strength, and temperature), the mixtures of protein and polysaccharide undergo associative or segregative phase separation. Usually, segregative phase separation appears when the protein and the polysaccharide are incompatible, carrying similar net charges and thus repelling one another. The formation of protein–polysaccharide complexes occurs when they attract each other through electrostatic interactions [8]. Other forces like hydrogen bonding or hydrophobic interactions may also play important roles in polyelectrolyte complex formation.

Over the years, growing interest in utilizing polyelectrolyte nanogels as drug delivery systems was observed due to many great advantages like nontoxicity, biocompatibility, good tolerance, and easy preparation techniques generally performed under mild conditions. In contrast to chemically cross-linked complexes, polyelectrolyte systems offer greater advantages for therapeutic substances through improving physicochemical characteristics like stability and dissolution [9]. Moreover, the pH-dependent property presented by these complexes allows controlling the release of several drugs, proteins, peptides, and genes [10].

In this regard, chitosan-based nanogels have demonstrated great potential and versatility as vehicles for loading biological macromolecules and delivering these materials to specific sites (e.g., skin, liver). Through its characteristics like pH sensitivity, mucoadhesion, biocompatibility, and low toxicity, chitosan contributes to great cellular membrane permeability and a facile paracellular and transcellular transport of the encapsulated drugs. Many research groups have thoroughly investigated how polyelectrolyte complexes useful for drug delivery applications can be formed by electrostatic interactions of chitosan with protein/polypeptide structures such as bovine serum albumin [11,12] or poly(aspartic acid) [13].

In addition, chitosan can be easily functionalized to obtain the desired targeted delivery of therapeutic agents. Chemical modification of chitosan results in the formation of several derivatives [14] with improved solubility in several biological media (e.g., intestinal media—*n*-trimethyl chitosan chloride [15]) or enhanced mucoadhesiveness (thiolated chitosan [16]). The primary amine (–NH_2_) groups of chitosan provide a reaction site for chemical modification, which can enhance the stability and drug encapsulation efficiency and thus open ways to various utilizations in pharmaceutical and medical fields [17]. Compared to other derivatives, chitosan functionalized with anhydrides like maleic anhydride [18] contributes not only to the solubilization of chitosan in aqueous solutions but also to the partial neutralization of primary amine groups that hinder the aggregation of chitosan macromolecular chains in pH levels from neutral to high. As the aggregation phenomenon is avoided at neutral pH, the polyelectrolyte nanogels based on chitosan modified with anhydrides prepared in physiological conditions afford tunable drug release and impact the pharmacokinetic profile of the loaded drug.

Bovine serum albumin (BSA) is a relatively large globular protein, with well-known structure and physicochemical properties, which possesses a heterogeneous distribution of charges. BSA also has a high solubility in water due to a large number of ionizable amino acids; with these characteristics, it has been used for the preparation of nanoparticles and as a drug carrier [19].

Although many research articles demonstrated the potential applications of chitosan modified with maleic anhydride in gene and drug delivery [20,21,22], there are no reports so far about nanogels as drug delivery systems fabricated via self-assembly from maleoyl chitosan—a derivative that contains amino and carboxyl groups and remains protonated even at neutral pH—and BSA.

In the present investigation, nanogels based on maleic anhydride functionalized chitosan and proteins such as BSA were obtained by the self-assembly technique, which is a simple green process involving low costs. The capacity of the two macromolecular compounds to interact at the molecular level and form polyelectrolyte complexes was assessed through diffraction light scattering (DLS) and ultraviolet–visible (UV-VIS) spectrophotometry measurements. The synergistic properties of the self-assembled nanogels were highlighted by the determination of their responsiveness to environmental pH and temperature changes, as well as their antibacterial properties after being loaded with amoxicillin (Amox), a broad-spectrum antibiotic [23] with low efficacy and bioavailability.

## 2. Materials and Methods

### 2.1. Materials

Chitosan (M_W_ = 80 kDa and deacetylation degree >75%), maleic anhydride (MA), acetic acid, methanol, acetone, and amoxicillin (Amox) were obtained from Sigma-Aldrich (Hamburg, Germany). Bovine serum albumin (BSA, Mw of ∼66,000 g/mol) was purchased from Fluka (Buchs, Switzerland) (minimum 98% electrophoresis, Fraction V).

### 2.2. Chitosan Functionalization with MA

Maleic anhydride chitosan derivative (MAC) was synthesized according to a reported procedure [24,25]. In brief, the functionalization of chitosan with MA was carried out in 5% acetic acid solution (chitosan/MA = 1/5 molar ratio) diluted with methanol by adding the anhydride dissolved in acetone under stirring in the chitosan solution. After 18 h, the resulted mixture was dialyzed in distilled water and freeze-dried. Static light scattering (SLS) measurements revealed a gravimetric molecular weight of 22.4 kDa for the chitosan derivative [25].

### 2.3. MAC/BSA Nanogel Preparation and Encapsulation of Amox

Prior to obtaining the self-assembled nanogels, stock solutions of MAC (0.125%, 0.25%, and 0.5% (*w/v*)) and BSA (0.125%, 0.25%, and 0.5% (*w/v*)) were prepared by dissolving appropriate amounts of each powder in the ultrapure Milli-Q water. The BSA/MAC nanogels were obtained by adding the aqueous solution of MAC to the BSA solution with stirring for 30 min. Polyelectrolyte complexes of different compositions were prepared by varying the ratio of BSA to MAC. Thus, BSA/MAC mass ratios studied were 100/1, 33/1, 20/1, 13.3/1, 10/1, 8/1, 7/1, 6/1, and 5/1. By increasing the maleic anhydride functionalized chitosan content above the 5/1 ratio, the formation of nanogels was no longer observed. To study the influence of temperature on the optimal conditions for self-assembly of nanogels, the same BSA/MAC ratios were prepared, stirred for 10 min, heated at 80 °C for 30 min, stirred for another 30 min, and cooled down at room temperature. In addition, the change of the order of the addition of compounds (BSA dripped in MAC solution) did not lead to the formation of nanogel systems (the solutions were transparent at 20 °C and 80 °C). After preparation, the mixtures were allowed to equilibrate for 24 h at 4 °C. For further characterization, the optimal system was lyophilized. Moreover, the optimal nanogel system was selected for drug loading. Thus, Amox was dissolved in the BSA solution to which the MAC was subsequently added. The mass ratio between BSA/MAC and Amox used for drug loading was 3:1.

The drug loading (DL) and the encapsulation efficiency (EE) percentages were calculated as follows: The resulting Amox-loaded nanogel suspension was centrifuged at 10,000 rpm for 15 min. The amount of unloaded drug was calculated from the absorbance at λ max of 231 nm of the supernatant (see Equations (1) and (2)).
(1)$$EE\;\left(\%\right)=\frac{initial\;amount\;of\;drug-amount\;of\;unloaded\;drug}{initial\;amount\;of\;drug\;taken\;for\;loading\;studies}$$
(2)$$DL\;\left(\%\right)=\frac{amount\;of\;drug\;in\;nanogels}{amount\;of\;polymer\;and\;drug\;taken\;for\;loading\;studies}$$

### 2.4. Turbidity Measurements of BSA/MAC Nanogels

The turbidity measurements of nanogel solutions based on BSA (fixed concentrations of 0.5%, 0.25%, and 0.125% *w/v*) and increasing amounts of maleic anhydride functionalized chitosan (0.5%, 0.25%, and 0.125% *w/v*) were recorded using a UV-Vis Spectrophotometer (Jenway 6305, Stone, Staffordshire, United Kingdom). The turbidity of the complex dispersions was measured at λ = 500 nm, where the polyelectrolytes do not absorb [26]. All recordings were made after preparing the samples in the same conditions as the ones presented in the above section and equilibrated for 24 h at 4 °C.

The system’s turbidity was expressed in arbitrary units as values of transmitted light (transmittance, T). T was expressed as the average of three measurements.

### 2.5. Dynamic Light Scattering (DLS) Measurements of BSA/MAC Nanogels

DLS measurements of the self-assembled nanogel complexes in aqueous solutions (particle size, polydispersity index (PDI)) were performed using Malvern Zetasizer Nano ZS instrument (Malvern Instruments, (Worcestershire, United Kingdom) equipped with a 4.0 mW He-Ne laser operating at 173° (backscatter mode) on 633 nm. The hydrodynamic diameters (D_h_) were calculated from diffusion coefficients using the Stokes–Einstein equation. Zeta potential measurements were also performed (Zeta-Sizer IV, Malvern Instruments, Worcestershire, United Kingdom)**.** For pH sensitivity investigation, the pH was adjusted by adding 0.1 N HCl and 0.1 N NH_4_OH solutions. Each sample was measured out in triplicate at 25 °C.

### 2.6. Fourier Transform Infrared (FT-IR) Analysis

The structure of BSA/MAC nanogels and precursors was analyzed with a Vertex Bruker spectrophotometer in transmittance mode using the ATR technique (Bruker, Ettlingen, Germany). Spectra were acquired at room temperature in a wavenumber range of 4000–400 cm^−1^ using 64 scans with a resolution accuracy of 4 cm^−1^.

### 2.7. Morphology of the BSA/MAC Nanogels

The morphologies of the BSA/MAC nanogels were evaluated by scanning electron microscopy (SEM) on a Quanta 200 instrument with an EDAX elemental analysis system operating with secondary electrons at 20 kV, under low vacuum mode (60–100 Pa) and LFD detector (FEI, Eindhoven, The Netherlands).

### 2.8. Determination of the In Vitro Antibacterial Activity of Loaded Nanogels Using the Disk Diffusion Assay

The antimicrobial activity of the optimized nanogel system was screened by disk diffusion assay against two different reference strains: *Escherichia coli* ATCC25922 (Gram-negative) and *Staphylococcus aureus* ATCC25923 (Gram-positive). Microbial suspensions were prepared with these cultures in sterile solution to obtain turbidity optically comparable to that of 0.5 McFarland standards.

Volumes of 0.5 mL from each inoculum were spread on Petri dishes. The sterilized paper disks were placed on the plates and an aliquot (10 μL) from samples was added. To evaluate the antimicrobial properties, the growth inhibition under standard conditions after 24 h of incubation at 36 ± 1 °C was measured. All tests were carried out in triplicate to verify the results. After incubation, the diameters of inhibition zones were measured by using Image J software (University of Wisconsin, Madison, WI, USA). 

The minimum inhibitory concentration (MIC) was determined by using the broth dilution method performed in 96-well microtiter plates. Briefly, bacterial culture grown to log phase was adjusted to 1 × 10^8^ cells/mL in Muller-Hinton (MH) Broth. Inoculants of 100 µL were mixed with 100 µL of serial dilutions of samples and were subsequently incubated at 37 °C for 24 h. The antibacterial activity was determined on the basis of turbidity by an EnVisions plate reader (PerkinElmer, Turku, Finland). The minimum bactericidal concentration (MBC) was determined by spreading 10 µL samples from wells on MH agar plates. In order to determine MIC/MBC values, the experiments were performed in triplicate.

## 3. Results and Discussion

Polyelectrolyte complexes are versatile formulations based on macromolecules with ionizable or ionic groups (polycation/polyanion) which exhibit unique properties due to their electrostatic interactions and flexibility. Thus, through complexation between BSA and maleic anhydride functionalized chitosan, the modified polysaccharide can act as a nanocarrier for antimicrobial drugs.

### 3.1. Mechanism of BSA/MAC Nanogel Formation

There are various parameters that have a significant effect on the self-assembly process; these parameters include the properties of the materials, such as the molecular weight of biopolymers, the total concentration of biopolymers, the mass mixing ratio of biopolymers, and their loading.

To highlight the self-assembly process between the two components, great attention was dedicated to assessing three main parameters, namely surface charge density (by measuring the zeta potential), particle size distribution, and turbidity [27], as a function of nanogel formation temperature and the ratio between polymers. Therefore, polyelectrolyte complexes of different compositions were prepared by varying the volume ratio of BSA to MAC, namely 100/1, 33/1, 20/1, 13.3/1, 10/1, 8/1, 7/1, 6/1, and 5/1, as presented in Table 1.

The surface electrical properties of biopolymers represent an important parameter for verifying the complexation capacity and stability of the formed nanogels. The use of ultraviolet–visible (UV-VIS) spectrophotometry to measure the turbidity of polymer mixtures can also illustrate the formation of the self-assembled complexes.

#### 3.1.1. Turbidity Measurements of Nanogel Solutions

In order to study the formation of colloidal polyelectrolyte complexes between BSA and MAC, the process was followed from the point of view of the mixing mass ratio between the two polymers. During nanogel formation, four types of phenomena were observed. The first one was a polyelectrolyte solution where one polymer was in excess; this was followed by an opalescent suspension, then flocculation, and finally transparent suspension (Figure 1). The transmittance of the nanogel solution was measured at the wavelength λ = 500 nm. For the system where 0.5% MAC was added dropwise to 0.5% BSA solution, from the ratio of 100:1 till 8:1 the transmittance decreased slowly, and the systems were all slightly opalescent. Further increase in the mass ratio led to the occurrence of flocculation until 7:1, at which point the system was opalescent and the transmittance began to rise again. This behavior evidenced the formation of macroscopic insoluble polyelectrolyte complexes, while the increase of transmittance can be explained by the fact that neither BSA nor MAC can absorb light at λ = 500 nm. Reaching a turbidity plateau (very low transmittance values) may indicate the electrostatic interaction between the negative and positive charges of the examined biopolymers resulting in the formation of the visible insoluble coacervated phase with specific coalescence behavior for all complexes coacervated.

The formation of self-assembled nanogels was also confirmed by the DLS measurements (D_h_, PDI, and zeta potential) (Figure 2).

When the complex nanogels were formed at 80 °C, lower values for transmittance were recorded (100:1 BSA/MAC ratio), which increased with the addition of MAC. A maximum of turbidity was recorded at the 13.3:1 BSA/MAC ratio, when the solution of polyelectrolyte nanogels became highly opaque. In this case, the conformational changes of BSA induced by heat contributed to the formation of aggregates as a result of the high charge density on the particle surface. Repulsion from the remaining negative charges led to high D_h_; these values were also confirmed by DLS measurements.

At a 13.3:1 BSA/MAC ratio, upon mixing the solutions of positively charged MAC and negatively charged BSA, the solutions begin to turn gradually turbid/opalescent and the transmittance decreased, indicating that the polyelectrolyte complexes were formed as expected due to the electrostatic interactions. The minimum transmittance value was detected when the weight ratio of BSA to MAC was 8:1, at 80 °C. As stated above, this point is considered a turbidity plateau where electrical equivalence (near isoelectric point) is achieved between the biopolymers, leading to coalescence owing to the newly formed complex coacervates which separate at the vial bottom. In the 7:1–5:1 mass ratio interval, the increased MAC mass content led to the condensation of denatured BSA molecules. When increasing the ratio of one polyelectrolyte to the other one, as already observed by other researchers, many coacervates typically reach a turbidity maximum, followed by a sudden loss in optical transmittance concretized in even a completely transparent polymer solution (5:1 ratio) [28,29,30].

A possible explanation can be formulated starting from the fact that with increasing the maleic anhydride functionalized chitosan to BSA ratio, an effect of “overcharging” of the coacervates can manifest, and the newly formed polyelectrolyte complexes become electrostatically repulsive and will be largely disassociated as the biopolymers carry a similar net charge [31,32].

Overall, the further increase in the maleic anhydride functionalized chitosan mass content leads to the decrease in the optical density of mixtures due to the formation of nonstoichiometric BSA/maleic anhydride functionalized chitosan complexes of variable composition.

Except for the flocculation of the system characterized by an abrupt drop of the optical density, the colloidal dispersions were stable, indicating that electrostatic stabilization prevented further coagulation. This stabilization might be ensured by an excess of binding of the major component (BSA), likely to form a stabilizing shell around the particles. When the concentration of polymers was decreased to 0.25%, aggregates were formed at 20 °C, with an abrupt drop in optical density at 8:1 BSA/MAC ratio due to neutralization of all charges leading to the coalescence of the aggregates. At 80 °C, the graphical appearance is similar, with the abrupt drop in optical density being observed at the 10:1 BSA/MAC ratio, where an electrical equivalence is achieved; these data are in agreement with the zeta potential measurements. In the case of the BSA/MAC systems prepared at a polymer concentration of 0.125%, at both temperatures, they were characterized by flocculation in the whole utilized range of mass ratios, in accordance with the zeta potential values which are closer to the isoelectric point over a large domain (from 10:1 to 5:1, excepting 7:1). This behavior is the consequence of the low charge density that is not sufficient to sustain the assembly at various ratios in the case of the nanogels formed at 0.5%. In the case of polymer solutions with 0.125% concentration, the electrostatic forces are too weak to stabilize the coacervates into more compact particles. Even the hydrophobic bonds resulting from BSA denaturation are not sufficient to sustain the self-assembly process; moreover, similar transmittance to mass ratio graphical appearances are observed for BSA/MAC systems prepared at 20 and 80 °C.

#### 3.1.2. Particle Size, PDI, and Zeta Potential Analysis by DLS

BSA is negatively charged in aqueous solutions since its isoelectric point is 4.7. On the other hand, MAC is a water-soluble derivative of chitosan with positive charge density at low pH. Nevertheless, the electrostatic attraction between BSA and MAC can be influenced by the ratio between the weights of the polymer partners. The experimental results evidenced the influence of the ratio of BSA to MAC upon the size and distribution of the nanogels. Thus, the zeta potential values of the BSA (0.5%) and MAC (0.5%) solutions were −37 and +41.6 mV, respectively, while the BSA/MAC systems had values ranging from −21 to +18 mV (Table 2, Figure 3). When BSA chains are in excess, the formed interpolymer complexes have groups with negative charges on the shell. As the ratio of maleic anhydride functionalized chitosan increases, macromolecular chains of MAC chain with free amino groups are exposed on the surface of BSA/MAC nanogels, leading to positive zeta potential values. The increase of the volume of added maleic anhydride functionalized chitosan led to the exponential increase of D_h_ of the particles up to a certain ratio (10/1), after which it gradually began to decrease (from 8:1 till 5:1) (Figure 3a). In the case of the ratios 100:1, 33:1, 20:1, 13.3:1, and 10:1, the particle solutions appeared slightly opaque to translucent as a result of the grouping of ion pairs of opposite charges of BSA with those of MAC. This behavior corresponds to the presence of quasineutral complexes without sufficient free negative groups to ensure particle stabilization. Therefore, the complexes formed rapidly aggregated within 24 h, with the aggregate settling at the base of the reaction vessel.

Denaturing proteins helps to stabilize the nanoparticulate systems by manifesting hydrophobic interactions and disulfide bonds that occur as a result of protein domain fragmentation [33]. Thus, nanogels based on BSA (0.5% *w/v*) and chitosan derivative (0.5% *w/v*) with the same ratios varying from 100:1 to 5:1 were formed after heating the mixtures for 30 min at 80 °C, without any chemical treatment. DLS measurements (Figure 3b) illustrate that particles with smaller dimensions and PDI are formed compared to the same nanogel systems obtained at room temperature (20 °C). Similar to nanogels prepared at 20 °C, their size increased with the addition of maleic anhydride functionalized chitosan (10:1 BSA/MAC ratio) (Table 2), after which it began to decrease, the nanogel solution becoming completely transparent. In this case (5:1 BSA/MAC ratios), were recorded values of approx. 347 nm for nanogel D_h_. This is probably because only a suitable ratio of BSA to MAC will favor the formation of self-assembled nanoparticles due to the opposite charges of the macromolecular chain partners, and further, an excess of BSA or MAC component will not favor the formation of nanogels.

Figure 3a,b shows the variation of D_h_ with the temperature used for the preparation of nanogels. Compared to 0.5BMAC7/1_20_, which had a particle diameter of 640 nm with PDI of 0.3, the same system obtained at 80 °C had a D_h_ of 210 nm with PDI of 0.169. The obtained results showed that the formation phases of the nanogels were based on the appearance of hydrophobic segments on the protein macromolecular chain, which stabilized the nanogels. It can be seen from Figure 3c that the zeta potential values of the BSA/MAC nanogels prepared at 80 °C were between −27.1 and +26.7 (with the achievement of neutrality and implicitly of the isoelectric zeta potential at a ratio of approx. 13.3:1). The values of the zeta potential increased proportionally with the addition of MAC solution. As compared with the values obtained for nanogels at room temperature, the zeta potential recorded for BSA/MAC systems obtained at 80 °C had higher values as a result of the cysteine residues/disulfide bridges exposed at the outer layer of the nanoparticles after heat denaturation of BSA molecules.

Thus, at 7:1 BSA/MAC ratio (80 °C), the particle suspension was opaque and more thermodynamically stable, the particles having dimensions below 300 nm and a PDI of 0.169, demonstrating the uniform distribution of the particle population. In this case, as a result of the neutralization of several charges, multiple conformational rearrangements of the macromolecular chains took place to form more compact particles and a surface with an even distribution of positive charges.

Taking into consideration the particle size, PDI, zeta potential, and transmittance measurements of nanogel solutions, the optimal BSA to MAC mass ratio for the preparation of nanogels was 7:1.

As a result, further investigations concerning the use of the BSA/MAC self-assembled systems as antimicrobial drug carriers were performed on nanogels synthesized with a BSA to MAC mass ratio of 7:1. The temperature of 80 °C was used for the preparation of the nanogels to obtain optimized size distribution and small PDI.

The influence of the concentration of the two polymers on the particle size was also investigated in order to find the optimum conditions for obtaining high-stability particulate systems, considering concentrations of 0.25% (*w/v*) and 0.125% (*w/v*).

BSA is a highly negative charged protein and requires highly positively charged polyelectrolyte for complete neutralization. In the case of nanogels based on 0.25% BSA and 0.25% MAC prepared at 20 °C, the increase in the volume of MAC neutralizes the negative charges on BSA to a greater extent and reduces the size of the complex (Figure 4a).

The tendency of particles to form aggregates was prominent in the case of BSA/MAC nanogels obtained at 80 °C, as can be observed from zeta potential and DLS measurements of the nanogels shown in Figure 4b. During heating of the BSA/MAC systems, protein unfolding occurs, followed by aggregation of unfolded protein molecules. It is likely that the rate of protein molecule aggregation is faster and hinders the interaction with the positively charged macromolecular chains of MAC as compared to the previous systems with BSA and MAC of 0.5% *w/v* concentration. This behavior is usually observed in the heating-induced denaturation of pure BSA in aqueous solutions and is determined by the multidomain structure of BSA [34]. According to the results of zeta potential measurements presented in Figure 4c, for both formation temperatures, the charges on nanogel surfaces were neutralized at 13.3:1 BSA/MAC ratio, confirming the approaching of the isoelectric point where nanogels are destabilized and have the tendency of aggregation.

Comparing the DLS results obtained for the systems based on BSA and MAC with concentrations of 0.125% (*w/v*) with those of the other two complex systems (0.5% and 0.25%), an increase of D_h_ is observed for all ratios, even for the systems obtained at 80 °C (Figure 5).

This phenomenon may be due to the low concentration and weak molecular mass of maleic anhydride functionalized chitosan (22.4 kDa), causing BSA to lose their attractive forces and detach from MAC macromolecular chains. Indeed, the transmittance to mass ratio and zeta potential curves of the systems show that the electrostatic screening is clearly lower. The zeta potential values of the nanogel dispersions underwent a drastic change, characterized by a charge switchover from negative to positive values, simultaneously with a sharp increase in D_h_ of the BSA/MAC complexes due to nanogel aggregation which ultimately led to the precipitation of the complexes. The isoelectric point occurs in the case of BSA/MAC nanogels prepared at 20 °C in the ratio range of 10:1 to 5:1 and at 7:1 for polymer systems synthesized at 80 °C.

Based on the obtained results, namely the smaller size and the adeptness for the formation, from the studied concentrations (0.125%, 0.25%, and 0.5%), the 0.5BMAC7/1_80_ system formed with 0.5% concentration of BSA and MAC solutions, at a mixing ratio of 7:1, was selected for further analyses and characterization.

For Amox-loaded nanogel, the loading efficacy with Amox drug was evaluated through DLS measurements, and the results indicated a monomodal distribution with slightly smaller D_h_ (198 nm) and reduced value of the zeta potential (+26) as compared with the unloaded nanogel system (Figure 6). These characteristics appear as a consequence of embedding the drug near its pKa value (4.8), which led to the occurrence of electrostatic attraction between the protonated free amino groups of MAC/BSA and the deprotonated carboxyl groups of Amox that stabilize the drug in the nanogel matrix. Additionally, it was found from the drug loading experiment that the EE was 95.9% and the DL was 31%, confirming the embedding of the drug in the nanogel network.

### 3.2. Structural Analysis of BSA/MAC Nanogels

The FT-IR spectrum of the maleic anhydride functionalized chitosan (MAC—Figure 7) presents intense peaks at 3257, 1703, and 1618 cm^−1^ which correspond to the stretching vibrations of the O–H and hydrogen bonds, the C=O stretching vibration of the carbonyl groups, and the stretching vibration of the unsaturated C=C bonds from maleoyl radicals, respectively. The characteristic absorption bands for BSA were registered at the following peaks: 3302 cm^−1^, indicating stretching vibration of O–H; 2959 and 2872 cm^−1^, indicating stretching vibration characteristics of the C–H bonds; 1661 cm^−1^, indicating amide I; and 1543 cm^−1^, indicating amide II. Some studies [35] reported that during the preparation of nanocarriers based on polycation/polyanion self-assembly, there is an increased possibility of physicochemical interactions, such as hydrogen bond formation and electrostatic interactions between the compounds involved in processes. In the spectrum corresponding to the 0.5BMAC7/1_80_ nanogel system, the absorption band characteristic of the O–H bond stretching vibration (BSA), but also the stretching vibration of the hydrogen bonds between O–H and N–H from MAC, moves from 3302 cm^−1^ to 3279 cm^−1^, the peak amplitude becoming larger in this case, indicating the formation of multiple hydrogen bonds. Moreover, in the nanogel spectrum, the characteristic peaks of amide I and amide II from BSA (main components in nanogels) were moved from 1661 to 1639 cm^−1^ and from 1543 to 1526 cm^−1^, respectively; these changes were attributed to the electrostatic interaction between the amphoteric polysaccharide and BSA. Thus, the FT-IR study points out the presence of characteristic bands of functional groups of both components, MAC and BSA, and also indicates the appearance of hydrogen bonds (between COOH and NH_2_ groups) and electrostatic interactions confirming the success of nanogel formation by self-assembly.

### 3.3. Morphological Analysis

The surface morphology of the nanogels based on BSA and MAC was evaluated by SEM analysis. Figure 8 presents SEM images of 0.5BMAC7/1_80_ nanogels, which are characterized by better size and formation ability, at different magnification orders. Irregular but predominantly spherical shapes and individual distributions indicative of the self-assembly process correspond to these samples. The chitosan-based systems consisted of numerous segregated or agglomerated particles packed closely together, which may be related to the dehydration process that occurs in the stage of preparing the samples for SEM analysis [36].

### 3.4. Investigation of pH and Temperature Sensitivity of BSA/MAC Nanogels

Comparative stability of the nanogels at varying pHs was studied using DLS measurements. The pH of nanogel solutions was adjusted to 4.0, 4.5, 4.7, 4.9, 5.2, 6.2, 7.2, 8.2, 8.5, and 9.1 using 0.1 M NH_4_OH or 0.1 M HCl solutions. D_h_, PDI, and zeta potential of the 0.5BMAC7/1_80_ nanogel system were measured. Figure 9 illustrates the evolution of the D_h_ of the nanogels with pH variation. Both maleic anhydride functionalized chitosan (MAC) and proteins such as BSA have surfaces that depend on the pH of the solution (positive charge at pH below the isoelectric point or pKa of the polysaccharide). Therefore, pH is considered a significant factor for the biopolymer complex formation. In general, the loading balance between proteins and polysaccharides depends on the ratios between biopolymers that can significantly influence the complex formation.

The results showed that the particle size decreased with the increase of pH until around 4.7, as a consequence of the self-assembly of the particles at a pH near the isoelectric point of BSA and above the pKa of maleic anhydride functionalized chitosan (4.2); a rise of the pH led to an increase of a particular size due not only to the deprotonation of the carboxyl group on the macromolecular chain of the polysaccharide but also to the amino acids like aspartic acid or glutamic contained in the BSA macromolecule. This means that from the pH = 4.7, the soluble complex begins to be established. Moreover, since the molar mass of the added polycation is low compared to that of the BSA, the size of the primary complex obtained in the 5.0–6.0 pH range is still small. The electrostatic repulsive forces between BSA and MAC occur due to the fact that both polymers are negatively charged over this pH range. Moreover, at pH 7.0, BSA/MAC nanogels were present in the form of relatively large particles, with a diameter greater than 500 nm, compared to those in the pH range of 5.0–6.0, which indicates the aggregation of polymers. Increasing the pH above 7.0 led to the occurrence of the shielding phenomenon of the deprotonated groups by the H^+^ ions from the NH_4_OH solution used to change the pH of the reaction medium.

The effect of the environmental temperature on particle size was also investigated by DLS measurements in the temperature range of 22–50 °C (Figure 10). An increase in D_h_ of BSA/MAC nanogels (7:1) prepared at 80 °C was observed throughout the whole investigated domain. The influence of temperature can result from changes in the configuration of the polymer (an entropic effect) or from shifts in the acid–base equilibria that control the dissociation and subsequent charging of weak polyelectrolyte complexes. This property may be useful in the case of the intracellular release of drugs, where swelling triggered by an increase in local temperature leads to a gradual release of the therapeutic agent.

### 3.5. Antimicrobial Tests

The antimicrobial activity of the 0.5BMAC7/1_80_ nanogels was determined by using the agar disk diffusion method. The compounds were added on the culture medium pre-inoculated with the microbial suspension, and the clear zone caused by the growth inhibition around the film disks after 24 h of incubation were measured. Nanogels without the antibiotic had no antibacterial activity, but in their combinations with Amox, they exhibited higher inhibition than 0.5BMAC7/1_80_ itself; a similar behavior was reported in the literature [37]. The 0.5BMAC7/1_80_ sample containing the antibiotic was more potent against *S. aureus* and less efficient in the case of *E. coli* (Table 3, Figure 11). Data on the diameters of the inhibition zones (mm) are presented in Table 3.

#### Determination of Minimum Inhibitory Concentration (MIC) and Minimum Bactericidal Concentration (MBC)

In order to determine MIC/MBC values, the experiments were performed in triplicate. MIC and MBC of the samples containing amoxicillin were 0.83 mg/mL for sample 0.5BMAC7/1_80_ against *S. aureus*, and the same value was established as MIC for *E. coli*. MBC against *E. coli* could not be estimated, probably requiring a higher concentration of amoxicillin within the samples. The high antimicrobial activity of these nanogels indicates good potential for applications in biomedical fields.

## 4. Conclusions

Self-assembled nanogels based on maleic anhydride functionalized chitosan and BSA were obtained and characterized to establish the optimal conditions of preparation. Through UV-VIS and DLS measurements, it was put into evidence that BSA/MAC nanogel systems prepared at 80 °C are stabilized by manifesting hydrophobic interactions and disulfide bonds occurring as a result of BSA domain fragmentation. Moreover, as a result of the neutralization of several charges, multiple conformational rearrangements of the macromolecular chains took place to form more compact particles and a surface with an even charge distribution. Study of the influence of concentration upon the nanogel formation shows that in the case of the nanoparticulate systems synthesized at 0.125% concentration of polymers, at two temperature values, namely 20 and 80 °C, a flocculation phenomenon appears in the whole utilized mass ratio range. The system prepared at a BSA to MAC volume ratio of 7:1 (0.5% *w/v*) was found to be more thermodynamically stable and was selected for Amox encapsulation and antimicrobial activity examination. After testing on *E. coli* and *S. aureus*, the system was demonstrated to be more efficient against *S. aureus*. The results showed that such systems can be considered as an attractive and promising formulation for antimicrobial drug delivery, thus generating a step forward towards skin engineering applications. Moreover, the utility of the BSA/MAC nanogels as delivery vehicles or components of composite materials would be dependent upon not only their size but also their tendency to swell or contract in response to environmental stimuli; thus, further investigations are required to understand the behavior of these nanogels in vitro and in vivo for potential use in biomedical applications.

## Figures and Tables

**Figure 1 polymers-12-02593-f001:**
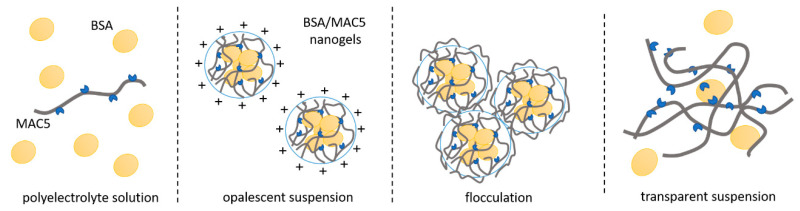
Types of phenomena that appear during BSA/MAC complexation.

**Figure 2 polymers-12-02593-f002:**
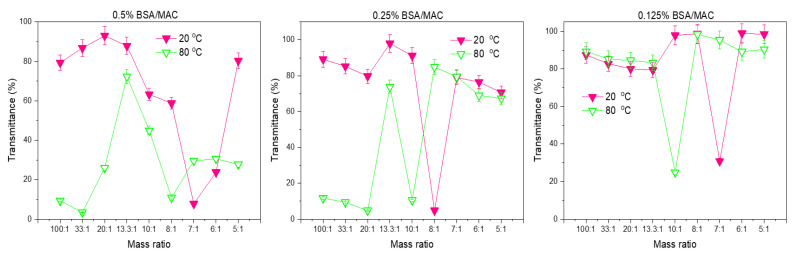
Variation of transmittance with BSA/MAC ratio at 20 and 80 °C.

**Figure 3 polymers-12-02593-f003:**
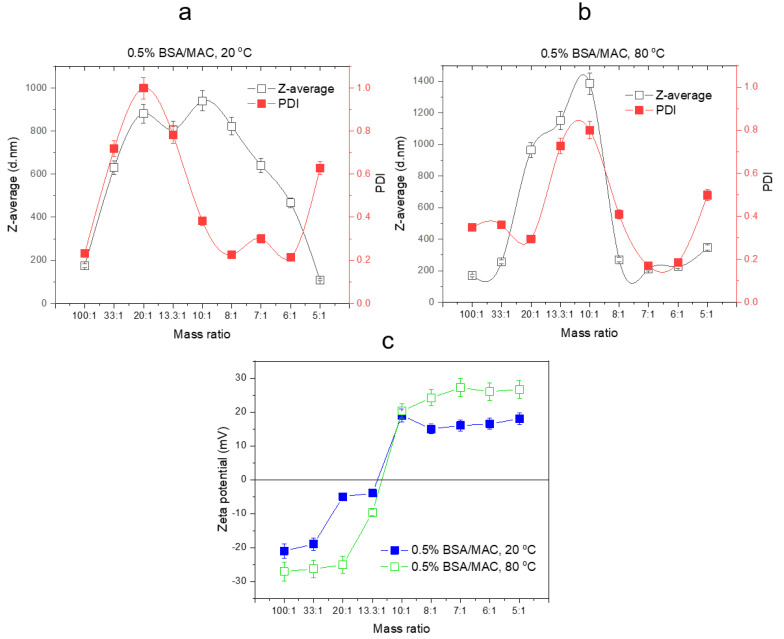
DLS (**a**,**b**) and zeta potential (**c**) evolution of the nanogels in relation to the synthesis conditions (BSA and MAC at 0.5% *w/v*).

**Figure 4 polymers-12-02593-f004:**
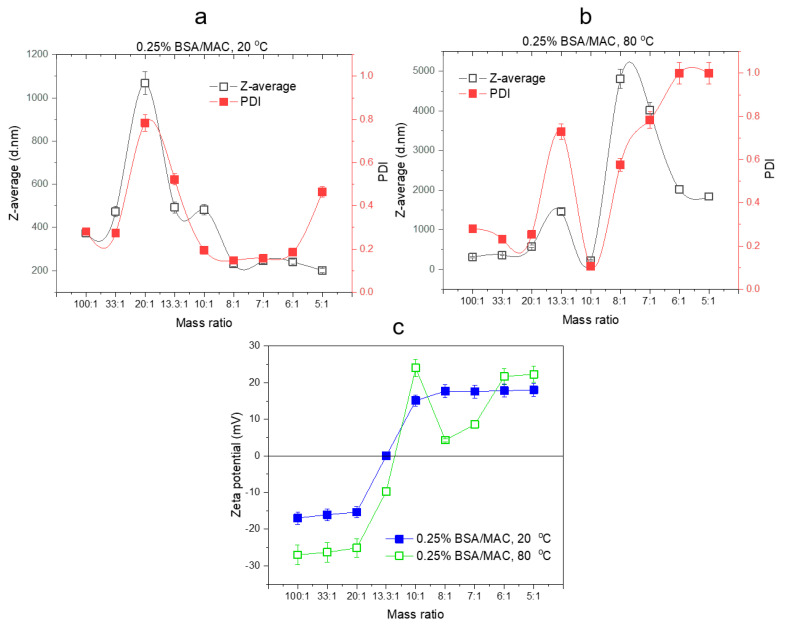
DLS (**a**,**b**) and zeta potential (**c**) results of nanogels synthesized with different weight ratios of BSA/MAC (0.25% *w/v*).

**Figure 5 polymers-12-02593-f005:**
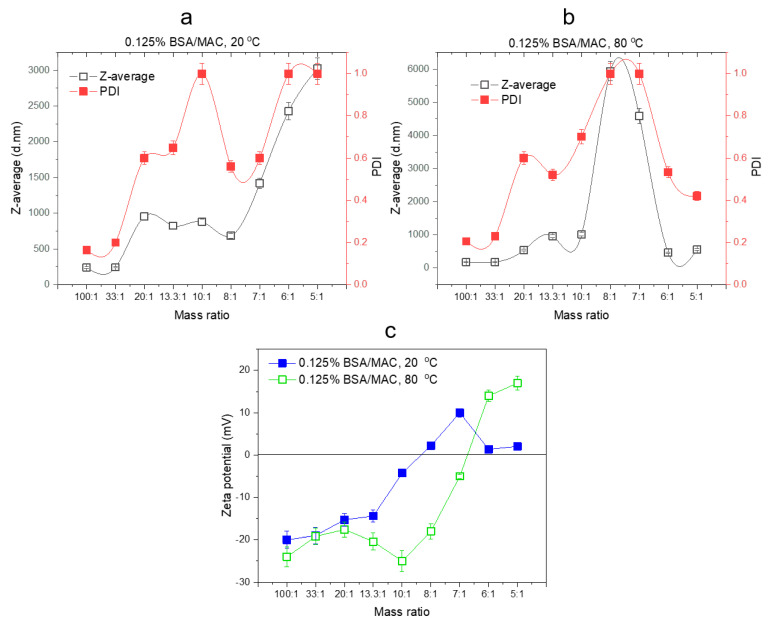
DLS (**a**,**b**) and zeta potential (**c**) results of nanogels synthesized with different weight ratios of BSA/MAC (0.125% *w/v*).

**Figure 6 polymers-12-02593-f006:**
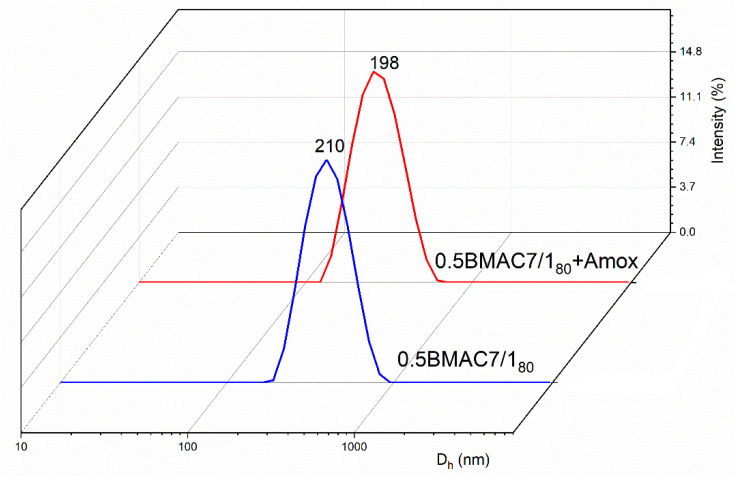
Size distribution of unloaded/amoxicillin (Amox)-loaded 0.5BMAC7/1_80_ nanogels.

**Figure 7 polymers-12-02593-f007:**
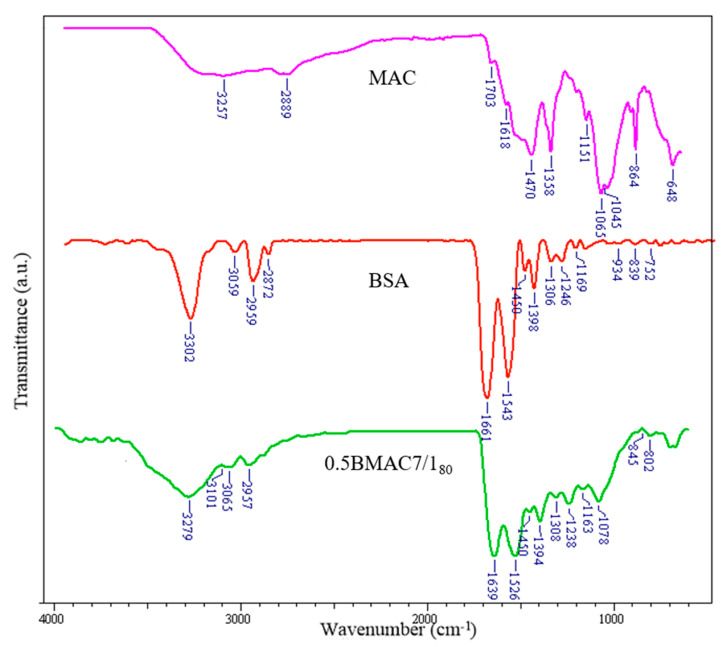
Fourier transform infrared (FT-IR) spectra of MAC, BSA, and 0.5BMAC7/1_80._

**Figure 8 polymers-12-02593-f008:**
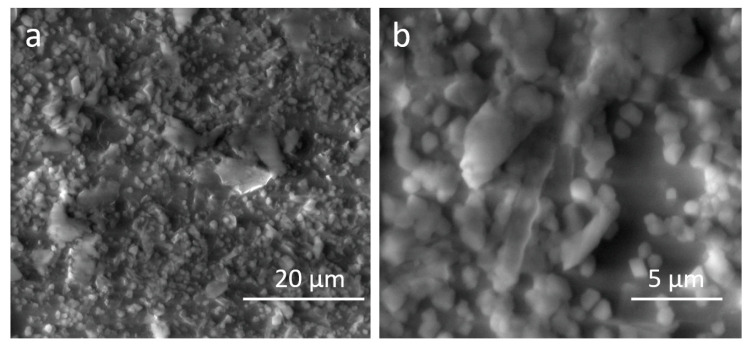
SEM images of 0.5BMAC7/1_80_ at different magnification orders: (**a**) 5000× and (**b**) 15,000×.

**Figure 9 polymers-12-02593-f009:**
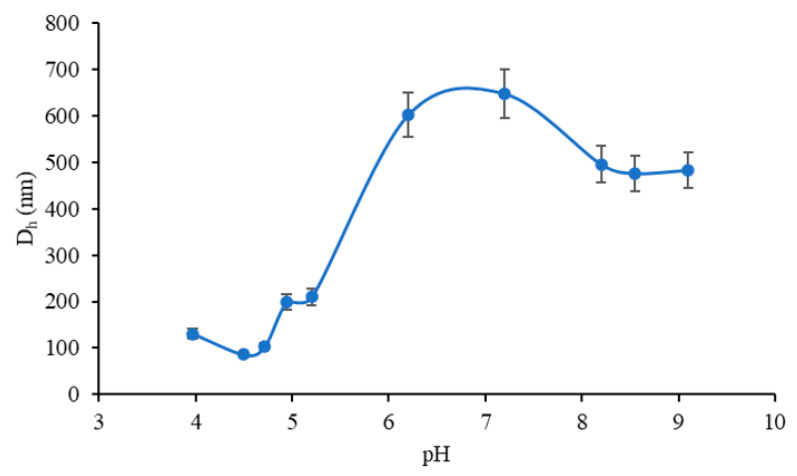
Variation of BSA/MAC nanogel D_h_ with pH.

**Figure 10 polymers-12-02593-f010:**
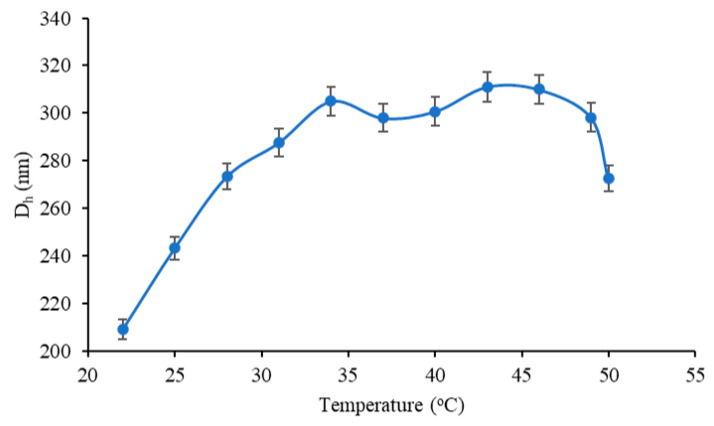
Variation of BSA/MAC nanogel D_h_ with temperature.

**Figure 11 polymers-12-02593-f011:**
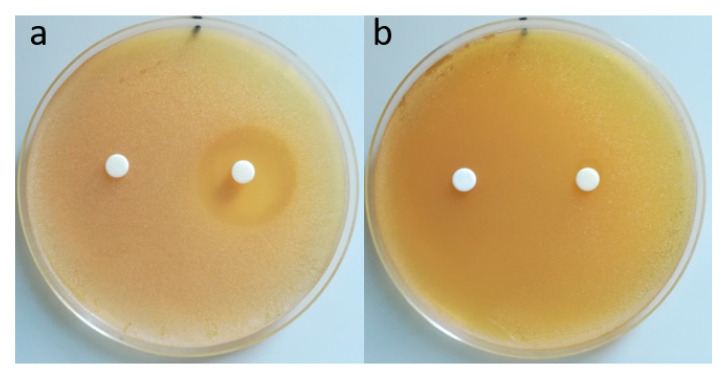
Antibacterial activity of nanogels 0.5BMAC7/1_80_ (left) without Amox and 0.5BMAC7/1_80_ with Amox (right) against *S. aureus* (**a**) and *E. coli* (**b**).

**Table 1 polymers-12-02593-t001:** Notation of bovine serum albumin (BSA)/maleic anhydride chitosan derivative (MAC nanogels formed at 0.5% polymer concentration.

Samples	BSA/MAC (0.5%) Weight Ratio
0.5BMAC100/1_20_ *	100:1
0.5BMAC33/1_20_	33:1
0.5BMAC20/1_20_	20:1
0.5BMAC13.3/1_20_	13.3:1
0.5BMAC10/1_20_	10:1
0.5BMAC8/1_20_	8:1
0.5BMAC7/1_20_	7:1
0.5BMAC6/1_20_	6:1
0.5BMAC5/1_20_	5:1
0.5BMAC100/1_80_	100:1
0.5BMAC33/1_80_	33:1
0.5BMAC20/1_80_	20:1
0.5BMAC13.3/1_80_	13.3:1
0.5BMAC10/1_80_	10:1
0.5BMAC8/1_80_	8:1
0.5BMAC7/1_80_	7:1
0.5BMAC6/1_80_	6:1
0.5BMAC5/1_80_	5:1

* Index represents the preparation temperature (°C).

**Table 2 polymers-12-02593-t002:** Characteristics of BSA and MAC nanogels obtained by diffraction light scattering (DLS) measurements.

Samples	Mass ratio BSA/MAC (0.5%)	20 °C			80 °C		
		D_h_ (nm)	PDI	Zeta Potential (mV)	D_h_ (nm)	PDI	Zeta Potential (mV)
0.5BMAC100/1	100:1	174	0.233	−21	168	0.348	−27.1
0.5BMAC33/1	33:1	6301	0.718	−19	255	0.359	−26.3
0.5BMAC20/1	20:1	881	0.998	−5	964	0.294	−25.1
0.5BMAC13.3/1	13.3:1	807	0.781	−4	1153	0.726	−9.7
0.5BMAC10/1	10:1	940	0.382	+19	1385	0.8	+20.3
0.5BMAC8/1	8:1	822	0.226	+15	270	0.408	+24.2
0.5BMAC7/1	7:1	640	0.3	+16	**210 ***	**0.169 ***	**+27.2 ***
0.5BMAC6/1	6:1	467	0.214	+16.5	226	0.185	+26
0.5BMAC5/1	5:1	110	0.628	+18	347	0.498	+27

* Index in the bold text represents DLS characteristics obtained for the most stable BSA/MAC nanogel system.

**Table 3 polymers-12-02593-t003:** Antimicrobial activities of the sample 0.5MAC7/_80_ unloaded and loaded with Amox against *S. aureus* and *E. coli*. MBC: minimum bactericidal concentration; MIC: minimum inhibitory concentration.

Strain	Inhibition Zone (mm)		MIC (mg/mL)		MBC (mg/mL)	
	0.5BMAC7/1_80_	0.5BMAC7/1_80_ + Amox	0.5BMAC7/1_80_	0.5BMAC7/1_80_ + Amox	0.5BMAC7/1_80_	0.5BMAC7/1_80_ + Amox
***S. aureus***	0	28.21 ± 0.09	Not tested	0.83	Not tested	0.83
***E. coli***	0	8.37 ± 0.12	Not tested	0.83	Not tested	-

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
