# Peer review of "Self-Assembled Nanocarriers Based on Modified Chitosan for Biomedical Applications: Preparation and Characterization"

_polymers, 2020, doi:10.3390/polym12112593_

Round 1
Reviewer 1 Report
The manuscript polymers-951586 titled “Self-Assembled Nanocarriers Based on Modified Chitosan for Biomedical Applications: Preparation and Characterization” was submitted to Polymers by Alina G. Rusu et al. to be published in the Special Issue "Crosslinked Nanogel Networks and Their Applications in Materials Science" that belongs to the section "Polymer Processing and Performance". The manuscript describes the preparation and characterization of maleic anhydride modified chitosan-BSA nanogels using different techniques and the evaluation of the nanogel as a carrier to release the encapsulated antibiotic amoxicilline. The subject of the work is suitable for the special issue and the aim of the work is relevant, but I consider the work needs mayor revisions to be accepted for publication. In the following points you will find a list of observations and suggestions that I consider indispensable for acceptance. After mayor revision by the authors the manuscript can be considered to be published in Polymers.
- In the Abstract. Line 14. Please correct to: maleic anhydride modified chitosan (MAC)
- All figures have a very low quality and very low resolution, graphics are unintelligible. They must be changed for high resolution images.
- The experiment you named UV–VIS spectral characterization of BSA/MAC nanogels, I do not consider it is a spectral characterization, it is a turbidimetry experiment at 500 nm. Should be corrected.
- In the UV–VIS turbidimetry assay of BSA/MAC nanogels, have you studied the kinetic of nanogel formation? After mixing BSA and MACs solutions, have you measured transmittance immediately or waited until stabilization? For how long? If so, it should be stated at the experimental. The time interval you measured Transmittance to build the graphics on figure 2 is relevant since nanogel formation must have a kinetic. Related to this, had the cell a magnetic stirring to ensure homogeneity of the solution?
- Since T was expressed as the average of three independent measurements, the graphics on figure 2 should include error bars. The same should be applied to the other experiments.
- I understand that the BSA and Mac solutions were made in pure water. What was the pH of such solutions? And after mixing and forming the nanogels? Any change? It is important since you are making conclusions based on charged systems.
- In line 226 you state “led to the condensation of BSA denaturated molecules” how do you know the BSA is denaturated?
- I consider the amox-encapsulated nanogel must be characterized as well (turbidimetry, DLS, SEM, FT-IR, etc) in order to account for the encapsulation of amoxicilline and validate the antibiotic activity assay. Since it may be a simple mixture of nanogel and a solution of amoxicilline.
- Also, the antibiotic activity of the potential encapsulated amoxicilline should be compared to the activity of a solution of free amoxicilline (same concentration, pH etc) to evaluate if the nanogel really improves the bioavailability as stated at the beginning of the manuscript.
Author Response
Answers to the Reviewers
The manuscript, entitled:
Self-Assembled Nanocarriers Based on Modified Chitosan for Biomedical Applications: Preparation and Characterization
authors:
Alina Gabriela Rusu, Aurica P. Chiriac, Loredana Elena Nita, Irina Rosca, Daniela Rusu, Iordana Neamtu
Firstly, we thank the Reviewers by helping us to improve our paper. The corrections were made and the manuscript was re-written accordingly with indications.
Comments from reviewer:
Reviewer 1
The manuscript polymers-951586 titled “Self-Assembled Nanocarriers Based on Modified Chitosan for Biomedical Applications: Preparation and Characterization” was submitted to Polymers by Alina G. Rusu et al. to be published in the Special Issue "Crosslinked Nanogel Networks and Their Applications in Materials Science" that belongs to the section "Polymer Processing and Performance". The manuscript describes the preparation and characterization of maleic anhydride modified chitosan-BSA nanogels using different techniques and the evaluation of the nanogel as a carrier to release the encapsulated antibiotic amoxicilline. The subject of the work is suitable for the special issue and the aim of the work is relevant, but I consider the work needs mayor revisions to be accepted for publication. In the following points you will find a list of observations and suggestions that I consider indispensable for acceptance. After mayor revision by the authors the manuscript can be considered to be published in Polymers.
- In the Abstract. Line 14. Please correct to: maleic anhydride modified chitosan (MAC)
R1: The modification was made as indicated.
- All figures have a very low quality and very low resolution, graphics are unintelligible. They must be changed for high resolution images.
R2: The quality of the figures was improved, but the problem still persist as the pdf conversion will be made directly from Office Word.
3.The experiment you named UV–VIS spectral characterization of BSA/MAC nanogels, I do not consider it is a spectral characterization, it is a turbidimetry experiment at 500 nm. Should be corrected.
R3: As reviewer indicated, the UV–VIS spectral characterization of BSA/MAC nanogels was removed and replaced with Turbidity measurements of BSA/MAC nanogels.
4.In the UV–VIS turbidimetry assay of BSA/MAC nanogels, have you studied the kinetic of nanogel formation? After mixing BSA and MACs solutions, have you measured transmittance immediately or waited until stabilization? For how long? If so, it should be stated at the experimental. The time interval you measured Transmittance to build the graphics on figure 2 is relevant since nanogel formation must have a kinetic. Related to this, had the cell a magnetic stirring to ensure homogeneity of the solution?
R4: The samples for turbidity measurements were prepared in the same conditions as the ones presented in Section 2.3, equilibrated for 24h at 4oC and gently stirred before analyzing. Simultaneously with the turbidity determinations were also realized DLS measurement to assess the stability of the samples.
New commentaries were inserted in the section dedicated to Turbidity measurements of BSA/MAC nanogels as follows:
“All recordings were made after preparing the samples in the same conditions as the ones presented in the above section and equilibrated for 24 hours at 4oC.”
5.Since T was expressed as the average of three independent measurements, the graphics on figure 2 should include error bars. The same should be applied to the other experiments.
R5: Modifications were made in accordance with the reviewer’s indications.
6.I understand that the BSA and Mac solutions were made in pure water. What was the pH of such solutions? And after mixing and forming the nanogels? Any change? It is important since you are making conclusions based on charged systems.
R6: The pH of the solutions was 5.6 for BSA and 4.0 for MAC5. All the commentaries made were in accordance with the samples pHs after mixing and varied between 5.4 and 5.6. The pH slightly decreased when a higher volume of MAC was added for samples preparation.
7.In line 226 you state “led to the condensation of BSA denaturated molecules” how do you know the BSA is denaturated?
R7: The observations regarding the denaturation of BSA when the temperature of nanogels preparation is raised above 40oC were made after consulting other papers that used a similar method to prepare nanogels based on BSA or other proteins. This method ensures the obtainment of nanogels with low dispersity index and small hydrodynamic diameter. In this regard, a list of articles was inserted below:
- Chen, N., Lin, L., Sun, W. and Zhao, M., 2014. Stable and pH-Sensitive Protein Nanogels Made by Self-Assembly of Heat Denatured Soy Protein. Journal of Agricultural and Food Chemistry, 62(39), pp.9553-9561.
- Yu, S., Hu, J., Pan, X., Yao, P. and Jiang, M., 2006. Stable and pH-Sensitive Nanogels Prepared by Self-Assembly of Chitosan and Ovalbumin. Langmuir, 22(6), pp.2754-2759.
- Hu, J., Yu, S. and Yao, P., 2007. Stable Amphoteric Nanogels Made of Ovalbumin and Ovotransferrin via Self-Assembly. Langmuir, 23(11), pp.6358-6364.
- Liu, K., Zheng, D., Zhao, J., Tao, Y., Wang, Y., He, J., Lei, J. and Xi, X., 2018. pH-Sensitive nanogels based on the electrostatic self-assembly of radionuclide131I labeled albumin and carboxymethyl cellulose for synergistic combined chemo-radioisotope therapy of cancer. Journal of Materials Chemistry B, 6(29), pp.4738-4746.
8.I consider the amox-encapsulated nanogel must be characterized as well (turbidimetry, DLS, SEM, FT-IR, etc) in order to account for the encapsulation of amoxicilline and validate the antibiotic activity assay. Since it may be a simple mixture of nanogel and a solution of amoxicilline.
R8. As the title of the article stated, this study aims to present the preparation and characterization of nanogels based on maleic anhydride modified chitosan and poly(aspartic acid). The results regarding the encapsulation and antimicrobial activity of the amoxicillin loaded nanogels are only preliminary studies regarding the potential medical application of the nanogels. More results will be included in another paper that highlights the feasibility of using these nanogels as drug delivery systems.
9.Also, the antibiotic activity of the potential encapsulated amoxicilline should be compared to the activity of a solution of free amoxicilline (same concentration, pH etc) to evaluate if the nanogel really improves the bioavailability as stated at the beginning of the manuscript.
R9: As it is known, the pharmacokinetic/pharmacodynamics activity of Amoxicillin were investigated for a long time as well as against S. Aureus and E. Coli, and the respective investigations constitute studies presented in specific papers (just 5 references in the following).
- Yao, Q., Gao, L., Xu, T., Chen, Y., Yang, X., Han, M., … Yang, Y. (2019). Amoxicillin Administration Regimen and Resistance Mechanisms of Staphylococcus aureus Established in Tissue Cage Infection Model. Frontiers in Microbiology, 10.doi:10.3389/fmicb.2019.01638
- Peric, M. (2003). Activity of nine oral agents against gram-positive and gram-negative bacteria encountered in community-acquired infections: Use of pharmacokinetic/pharmacodynamic breakpoints in the comparative assessment of beta-lactam and macrolide antimicrobial agents. Clinical Therapeutics, 25(1), 169–177.doi:10.1016/s0149-2918(03)90021-x
- Bantar, C., Nicola, F., Arenoso, H. J., Galas, M., Soria, L., Dana, D., … Jasovich, A. (1999). Pharmacokinetics and Pharmacodynamics of Amoxicillin-Sulbactam, a Novel Aminopenicillin–β-Lactamase Inhibitor Combination, against Escherichia coli. Antimicrobial Agents and Chemotherapy, 43(6), 1503–1504.doi:10.1128/aac.43.6.1503
- Skarp, K.-P., Shams, A., Montelin, H., Lagerbäck, P., & Tängdén, T. (2018). Synergistic and bactericidal activities of mecillinam, amoxicillin and clavulanic acid combinations against ESBL-producing Escherichia coli in 24-h time-kill experiments. International Journal of Antimicrobial Agents.doi:10.1016/j.ijantimicag.2018.09.011
- Lees, P., Pelligand, L., Illambas, J., Potter, T., Lacroix, M., Rycroft, A., & Toutain, P.-L. (2015). Pharmacokinetic/pharmacodynamic integration and modelling of amoxicillin for the calf pathogensMannheimia haemolyticaandPasteurella multocida. Journal of Veterinary Pharmacology and Therapeutics, 38(5), 457–470.doi:10.1111/jvp.12207
More than that the mechanisms of action of amoxicillin against S. aureus are unclear.
- Yao, Q., Gao, L., Xu, T., Chen, Y., Yang, X., Han, M.,Yang, Y. (2019). Amoxicillin Administration Regimen and Resistance Mechanisms of Staphylococcus aureus Established in Tissue Cage Infection Model. Frontiers in Microbiology, 10. doi:10.3389/fmicb.2019.01638
We have not yet set out to study the synergies that such system can bring to the antimicrobial effect. We considered sufficient for the study the highlighting of the antimicrobial character of the presented system. However, we thank Reviewer for the suggestion, which may be useful for the further development of our study.

Reviewer 2 Report
Authors describe a way to a protein–polysaccharide system for the preparation of nanogels based on modified chitosan derivatives (MAC) and bovine serum albumin (BSA), study the respective properties in order to optimize the best composition and additionally study the potential use as drug carriers.
The subject is interesting, actual and adequate for the journal. The structure of the manuscript is well done. However some things must be improved and corrected.
The discussion is quite superficial, more like a description of the work done with very few references to other work and not comparing the obtained results with literature ones. As examples I include only 5 DOI references closely related:
DOI: 10.3390/membranes9120163
DOI: 10.1007/s00289-020-03335-9
DOI: 10.1007/s10904-019-01421-8
DOI: 10.1016/j.msec.2016.05.121
DOI: 10.1016/j.ijpharm.2010.05.034
The quality of the Figures must be improved. Some of them are quite difficult to understand (namely Figures 2, 3 and 4).
In Figures 2, 3 and 4, maybe it will be a good idea to have the same PDI scale on a and b images for an easy and immediate comparison.
The paragraph between lines 178 and 181 is not very clear. It should be improved.
The English is very acceptable, but it should be revised. Eg. In line 213 it should be “A maximum of turbidity…”
I think, the paper is suitable to be accepted, but requires some changes.
Author Response
Answers to the Reviewers
The manuscript, entitled:
Self-Assembled Nanocarriers Based on Modified Chitosan for Biomedical Applications: Preparation and Characterization
authors:
Alina Gabriela Rusu, Aurica P. Chiriac, Loredana Elena Nita, Irina Rosca, Daniela Rusu, Iordana Neamtu
Firstly, we thank the Reviewers by helping us to improve our paper. The corrections were made and the manuscript was re-written accordingly with indications.
Comments from reviewer:
Reviewer 2
Authors describe a way to a protein–polysaccharide system for the preparation of nanogels based on modified chitosan derivatives (MAC) and bovine serum albumin (BSA), study the respective properties in order to optimize the best composition and additionally study the potential use as drug carriers.
The subject is interesting, actual and adequate for the journal. The structure of the manuscript is well done. However some things must be improved and corrected.
The discussion is quite superficial, more like a description of the work done with very few references to other work and not comparing the obtained results with literature ones. As examples I include only 5 DOI references closely related:
DOI: 10.3390/membranes9120163
Casimiro, Ferreira, Leal, Pereira and Monteiro, 2019. Ionizing Radiation for Preparation and Functionalization of Membranes and Their Biomedical and Environmental Applications. Membranes, 9(12), p.163.
DOI: 10.1007/s00289-020-03335-9
Nikfarjam, M. and Kokabi, M., 2020. Chitosan/laponite nanocomposite nanogels as a potential drug delivery system. Polymer Bulletin
DOI: 10.1007/s10904-019-01421-8
Ma, Y., Song, Y., Ma, F. and Chen, G., 2019. A Potential Polymeric Nanogel System for Effective Delivery of Chlorogenic Acid to Target Collagen-Induced Arthritis. Journal of Inorganic and Organometallic Polymers and Materials, 30(7), pp.2356-2365.
DOI: 10.1016/j.msec.2016.05.121
Debele, T., Mekuria, S. and Tsai, H., 2016. Polysaccharide based nanogels in the drug delivery system: Application as the carrier of pharmaceutical agents. Materials Science and Engineering: C, 68, pp.964-981.
DOI: 10.1016/j.ijpharm.2010.05.034
Casimiro, M., Gil, M. and Leal, J., 2010. Suitability of gamma irradiated chitosan based membranes as matrix in drug release system. International Journal of Pharmaceutics, 395(1-2), pp.142-146.
R1: I thank the reviewer for the indications. However, some of the articles listed above as examples of other studies are not quite suitable for comparison with the nanogels prepares in this study: e.g: DOI: 10.3390/membranes9120163 is a review that discuss the utilization of ionizing radiation for membrane preparation and functionalization of polymer-based membranes for biomedical and environmental applications. Also, the nanogels system based on chitosan/laponite and obtained through ionic gelation with TPP which is discussed in DOI: 10.1007/s00289-020-03335-9 (Nikfarjam, M. and Kokabi, M., 2020. Chitosan/laponite nanocomposite nanogels as a potential drug delivery system. Polymer Bulletin) is a very distinct system as compared to the self-assembled maleoyl-chitosan/BSA nanogels prepared through electrostatic interactions. Additionally, where was the case, observations were made for comparison and more suitable articles were cited (see below an excerpt from the current manuscript:
“3.1.1. Turbidity measurements of nanogels solutions
In the 7:1-5:1 mass ratio interval, the increased MAC mass content led to the condensation of BSA denaturated molecules. When increasing one polyelectrolyte ratio reported to the other one, as it was already observed by other researchers, many coacervates typically reach a turbidity maximum, followed by a sudden loss in optical transmittance concretized in even a complete transparent polymer solution (5:1 ratio) [28–30].”
“3.5. Antimicrobial tests
Nanogels without antibiotic had no antibacterial activity, but in their combinations with Amox, it gives higher inhibition than 0.5BMAC7/180 itself; a similar behavior was reported in literature [37]. 0.5BMAC7/180 sample containing antibiotic was more potent against S. aureus and less efficient in case of E. coli (Table 3, Figure 10). Data on the diameters of the inhibition zones (mm) are presented in Table 3).”
2.The quality of the Figures must be improved. Some of them are quite difficult to understand (namely Figures 2, 3 and 4).
R2: The quality of the figures was improved.
3.In Figures 2, 3 and 4, maybe it will be a good idea to have the same PDI scale on a and b images for an easy and immediate comparison.
R3: Modifications were made in accordance with reviewer indications.
4.The paragraph between lines 178 and 181 is not very clear. It should be improved.
R4: The paragraph was corrected.
5.The English is very acceptable, but it should be revised. Eg. In line 213 it should be “A maximum of turbidity…”
R5: The manuscript language was improved.
I think, the paper is suitable to be accepted, but requires some changes.

Reviewer 3 Report
Authors team (Alina Gabriela Rusu, Aurica P. Chiriac, Loredana Elena Nita, Irina Rosca, Daniela Rusu, Iordana Neamtu) in manuscript polymers-951586, entitled “Self-Assembled Nanocarriers Based on Modified Chitosan for Biomedical Applications: Preparation and Characterization” are presenting nanogels based on maleic anhydride-functionalized chitosan and bovine serum albumin, obtained by self-assembly technique. The capacity of two macromolecular compounds to interact and to form polyelectrolyte complexes was evaluated by using DLS and UV-VIS methods. The synergistic properties of obtained nanogels were determined by their responsiveness to pH and temperature changes, as well as their antibacterial properties after loading with amoxicillin.
In my opinion, this manuscript provides useful results and it can be published in journal "Polymers" after required improvements:
- Is BCA an anionic polymer or protein? It is needed to rewrite sentence: "In the present investigation nanogels based on maleic anhydride-functionalized chitosan and anionic polymers such as BSA were obtained by the self-assembly technique, which is a simple green process involving low costs."
- According described method of synthesis in references 24 and 25, in part 1. Materials it is needed to add all used chemicals.
- Sentence: "Cs derivative (MAC) was synthesized according to a reported procedure [24,25]." It is needed to rewrite for better explanation of obtained product with full names of reactants and obtained product (maleic anhydride functionalized chitosan - abbreviated as MAC, explained in lines 391-392).
- Figures 2, 3, 4, 5, 7 are very low quality. It needs to be readable and with better quality. It is necessary for authors to correct, so it will be understandable for the readers analysis.
- Please, it is needed to update captions for Figures 2 with addition of presented sample names. Also, in Figures 3, 4, 5, 6 captions are missing explanation for a), b) and c) parts. It is needed to add information about which samples was used with defined polymers concentration. In all figures captions need to be updated with information about presented nanogel samples, e.g. 0.5BMAC7/180. Figure 7 is needs to be magnified to improve visibility and and put some more explanation about micrographs of nanogels samples defined as: a) and b) parts or left and right.
- Lines 368-371: "Also, in the nanogels spectrum, the characteristic peaks of BSA (mainly component in nanogels) from 1661cm-1 and 1543 cm-1 (amide I and amide II) are significantly lower and are moved to 1639cm-1 and 1526 cm-1, respectively, changes attributed to the electrostatic interaction between the amphoteric polysaccharide and BSA." It will be better to rewrite this sentence without repetition of wave numbers for amide I and amide II. It is needed to add calculated value by how much units absorption maxima of amide I and amide II have moved in relation to BCA and MAC.
- Whether the movements indicate weak or strong interactions?
- How authors explain loss of band- shoulder at 1703 cm-1 from the carbonyl groups in the MAC? It will be useful to add explanation which groups of BCA and MAC can interact and form hydrogen bonds?
- Please, it is needed to provide full names of acronyms on first appearance in the main manuscript text and figure caption (not only in abstract) (e.g.: for Dh). One example: Figure 6. FT-IR spectra of maleic anhydride functionalized chitosan (MAC), bovine serum albumin (BSA) and nanogel sample 0.5BMAC7/180.
- It is needed to replace all appearances of term "modified chitosan" to "maleic anhydride functionalized chitosan" for better explanation of the experiment and results (e.g. lines 129, 234, 259, 445).
- Please, it is needed to write italicized bacterial names: aureus and E. coli (Table 3. line 431) and full name for amoxycilin.
- Please, avoid 1st person plural and rewrite sentences to the 3rd person plural (line 457).
- Please, it is needed to harmonize all cited literature according to the Journal`s Instruction for authors (e.g. ref. 33, line 550; abbreviated journal names (e.g. line 510, 512, 517, 532, 541...), DOI numbers when is possible).
Author Response
Answers to the Reviewers
The manuscript, entitled:
Self-Assembled Nanocarriers Based on Modified Chitosan for Biomedical Applications: Preparation and Characterization
authors:
Alina Gabriela Rusu, Aurica P. Chiriac, Loredana Elena Nita, Irina Rosca, Daniela Rusu, Iordana Neamtu
Firstly, we thank the Reviewers by helping us to improve our paper. The corrections were made and the manuscript was re-written accordingly with indications.
Comments from reviewer:
Reviewer 3
Authors team (Alina Gabriela Rusu, Aurica P. Chiriac, Loredana Elena Nita, Irina Rosca, Daniela Rusu, Iordana Neamtu) in manuscript polymers-951586, entitled “Self-Assembled Nanocarriers Based on Modified Chitosan for Biomedical Applications: Preparation and Characterization” are presenting nanogels based on maleic anhydride-functionalized chitosan and bovine serum albumin, obtained by self-assembly technique. The capacity of two macromolecular compounds to interact and to form polyelectrolyte complexes was evaluated by using DLS and UV-VIS methods. The synergistic properties of obtained nanogels were determined by their responsiveness to pH and temperature changes, as well as their antibacterial properties after loading with amoxicillin.
In my opinion, this manuscript provides useful results and it can be published in journal "Polymers" after required improvements:
1.Is BCA an anionic polymer or protein? It is needed to rewrite sentence: "In the present investigation nanogels based on maleic anhydride-functionalized chitosan and anionic polymers such as BSA were obtained by the self-assembly technique, which is a simple green process involving low costs."
R1: The phrase was modified in accordance with the observations of the reviewer.
- According described method of synthesis in references 24 and 25, in part 1. Materialsit is needed to add all used chemicals.
R2: The Materials section was completed with all the solvents (acetic acid, methanol, acetone) used for chitosan modification.
- Sentence: "Cs derivative (MAC) was synthesized according to a reported procedure [24,25]." It is needed to rewrite for better explanation of obtained product with full names of reactants and obtained product (maleic anhydride functionalized chitosan - abbreviated as MAC, explained in lines 391-392).
R3: Modifications were made accordingly in the manuscript.
4.Figures 2, 3, 4, 5, 7 are very low quality. It needs to be readable and with better quality. It is necessary for authors to correct, so it will be understandable for the readers analysis.
R4: The quality of the images was improved.
5.Please, it is needed to update captions for Figures 2 with addition of presented sample names. Also, in Figures 3, 4, 5, 6 captions are missing explanation for a), b) and c) parts. It is needed to add information about which samples was used with defined polymers concentration. In all figures captions need to be updated with information about presented nanogel samples, e.g. 0.5BMAC7/180. Figure 7 is needs to be magnified to improve visibility and and put some more explanation about micrographs of nanogels samples defined as: a) and b) parts or left and right.
R5: The image quality has been improved and now the reviewer can see more clearly that Figure 2 shows the variation of the transmittance recorded for each sample at BSA / MAC ratio from 100: 1 to 5: 1 (Table 1) at various concentrations (0, 5%, 0.25% and 0.125%) and prepared at 20 oC or 80 oC.
The captions of Figures 3, 4, 5, 6 were improved. Moreover, in all mentioned Figures were presented results obtained for all prepared system with ratios starting from 100:1, 33:1, 20:1, 13.3:1, 10:1, 8:1, 7:1, 6:1 and 5:1.
The Figure 7 was magnified to improve the visibility and the figure caption was completed with new indications.
6.Lines 368-371: "Also, in the nanogels spectrum, the characteristic peaks of BSA (mainly component in nanogels) from 1661cm-1 and 1543 cm-1 (amide I and amide II) are significantly lower and are moved to 1639cm-1 and 1526 cm-1, respectively, changes attributed to the electrostatic interaction between the amphoteric polysaccharide and BSA." It will be better to rewrite this sentence without repetition of wave numbers for amide I and amide II. It is needed to add calculated value by how much units absorption maxima of amide I and amide II have moved in relation to BCA and MAC.
R6: Changes were made in the indicated paragraph as the reviewer indicated.
“Also, in the nanogels spectrum, the characteristic peaks of amide I and amide II from BSA (mainly component in nanogels) were moved from 1661 to 1639 cm-1 and from 1543 to 1526 cm-1, respectively, changes attributed to the electrostatic interaction between the amphoteric polysaccharide and BSA.”
A calculation will not rend an exact observation regarding the movement of the bands as ATR is much less sensitive compared to transmission experiment through KBr. Measurement with ATR is highly dependent on the material (diamond) of the crystal.
7.Whether the movements indicate weak or strong interactions?
R7: In general, movements of absorption to higher values of frequency are an indicator of strong interactions. In this case, as the values are decreasing, probably the interactions are weak. The simple occurrence of the physical interactions between the specific groups from BSA and MAC are influencing the shifting of bands that is visible in the FT-IR spectrum of 0.5BMAC7/180 system.
8.How authors explain loss of band- shoulder at 1703 cm-1 from the carbonyl groups in the MAC? It will be useful to add explanation which groups of BCA and MAC can interact and form hydrogen bonds?
R8: Most probably the band have not disappeared, but as the ratio of MAC reported to BSA is lower and the ATR technique is much less sensitive, it is not present in the spectrum of nanogel system.
9.Please, it is needed to provide full names of acronyms on first appearance in the main manuscript text and figure caption (not only in abstract) (e.g.: for Dh). One example: Figure 6. FT-IR spectra of maleic anhydride functionalized chitosan (MAC), bovine serum albumin (BSA) and nanogel sample 0.5BMAC7/180.
R9: The changes were made in accordance with the indications of the reviewer.
10.It is needed to replace all appearances of term "modified chitosan" to "maleic anhydride functionalized chitosan" for better explanation of the experiment and results (e.g. lines 129, 234, 259, 445).
R10: The term "modified chitosan" was replaced to "maleic anhydride functionalized chitosan” in the whole manuscript.
11.Please, it is needed to write italicized bacterial names: aureus and E. coli (Table 3. line 431) and full name for amoxycilin.
R11: The corrections were made in the manuscript for S. aureus and E. coli (Table 3). Also, the figure corresponding to S. aureus and E. coli were noted with a and b, respectively.
Regarding the addition of the full name of amoxicillin, the term was initially specified in the Materials section, and the abbreviation was further used in the text for the ease of presentation of the prepared complex.
12.Please, avoid 1st person plural and rewrite sentences to the 3rd person plural (line 457).
R12: The modification was made accordingly.
13.Please, it is needed to harmonize all cited literature according to the Journal`s Instruction for authors (e.g. ref. 33, line 550; abbreviated journal names (e.g. line 510, 512, 517, 532, 541...), DOI numbers when is possible).
R13: All the references were corrected.

Round 2
Reviewer 1 Report
After carefully reading the revised version of the manuscript “polymers-951586” submitted by A. G. Rusu et al. and the answers and revision I consider the authors have improved and clarified the preparation and characterization section of their work, related to questions 1-7.
Nevertheless, i consider the points 8 and 9 were not correctly assessed for the following reasons.
R8. As the title of the article stated, this study aims to present the preparation and characterization of nanogels based on maleic anhydride modified chitosan and poly(aspartic acid). The results regarding the encapsulation and antimicrobial activity of the amoxicillin loaded nanogels are only preliminary studies regarding the potential medical application of the nanogels. More results will be included in another paper that highlights the feasibility of using these nanogels as drug delivery systems.
The experiment performed as a preliminary study only can led to the conclusion that the “nanogel system” doesn’t interfere with the activity of amoxicillin. I wrote nanogel in quotes because the formation of the gel nanoparticles in the presence of amoxicillin was not experimentally confirmed. And from my point of view is not correct to suppose that the antibiotic was encapsulated, if there is no experimental evidence even of the formation of nanogels.
In several parts of the manuscript the authors mention “loaded nanogels” but there are not experiment that demonstrate the nanogel is loaded with amoxicillin, the solution is loaded with amoxicillin and it may just be dissolved in the water and, so, it will be bioactive.
If the experiment named “In vitro antibacterial activity of loaded nanogels using the disk diffusion assay” wants to be used as a preliminary result A DLS or SEM image of the “loaded nanogel particles” , for example, would be necessary.
9.Also, the antibiotic activity of the potential encapsulated amoxicilline should be compared to the activity of a solution of free amoxicilline (same concentration, pH etc) to evaluate if the nanogel really improves the bioavailability as stated at the beginning of the manuscript.
R9: As it is known, the pharmacokinetic/pharmacodynamics activity of Amoxicillin were investigated for a long time as well as against S. Aureus and E. Coli, and the respective investigations constitute studies presented in specific papers (just 5 references in the following).
I think the authors did not understand my comment. I know amoxicillin is a well-known, studied and effective antibiotic. My question has nothing to do with its mode of action. I understand this is a preliminary experiment but, In any antibiotic activity assay (or any other biological activity assay), the use of a positive control (if it axists) should be used as routine, in this case a solution of pure amoxicillin. If not used, why determine the MIC if there is nothing to compare it and to validate the assay?.
From my point of view, the synthesis of amoxicillin loaded nanogels and the preliminary experiment on antibiotic activity are incomplete, and the paper cannot be published as it is unless the previous points are assessed.
Author Response
Reviewer 1
The experiment performed as a preliminary study only can led to the conclusion that the “nanogel system” doesn’t interfere with the activity of amoxicillin. I wrote nanogel in quotes because the formation of the gel nanoparticles in the presence of amoxicillin was not experimentally confirmed. And from my point of view is not correct to suppose that the antibiotic was encapsulated, if there is no experimental evidence even of the formation of nanogels.
In several parts of the manuscript the authors mention “loaded nanogels” but there are not experiment that demonstrate the nanogel is loaded with amoxicillin, the solution is loaded with amoxicillin and it may just be dissolved in the water and, so, it will be bioactive.
If the experiment named “In vitro antibacterial activity of loaded nanogels using the disk diffusion assay” wants to be used as a preliminary result A DLS or SEM image of the “loaded nanogel particles” , for example, would be necessary.
R: To answer the reviewer’s doubts regarding the use of “loaded nanogels”, a graph representing the unloaded/loaded nanogels size distribution was inserted in the manuscript. Also, the data regarding the encapsulation efficiency was added to confirm that indeed the encapsulation was achieved.
I think the authors did not understand my comment. I know amoxicillin is a well-known, studied and effective antibiotic. My question has nothing to do with its mode of action. I understand this is a preliminary experiment but, In any antibiotic activity assay (or any other biological activity assay), the use of a positive control (if it axists) should be used as routine, in this case a solution of pure amoxicillin. If not used, why determine the MIC if there is nothing to compare it and to validate the assay?.
From my point of view, the synthesis of amoxicillin loaded nanogels and the preliminary experiment on antibiotic activity are incomplete, and the paper cannot be published as it is unless the previous points are assessed.
R: The authors understood very well the question of the reviewer. But in the case of this type of system where the drug interacts also with the nanogel compounds making the system more stable, the antimicrobial assay needs to be done for a long period of time in order to justify the necessity of a positive control (pristine drug without carriers). Moreover, as a positive control, it was used the unloaded nanogel solution that resulted in no antimicrobial activity (as it was expected). In this study it was never mentioned that the unloaded nanogels will have antimicrobial properties. It is obviously that the antimicrobial activity of the new amoxicillin-loaded nanogels will be given by the drug and released in a controlled manner as compared with the drug solution. From our point of view those data are enough to sustain the affirmations made in the section dedicated to preliminary antimicrobial activity.
But to answer the complains of the reviewer, the pristine amoxicillin was tested. As the reviewer can see in the images and table inserted below, the drug had antimicrobial activity against both microorganisms and better inhibitions zones as compared to the amoxicillin loaded nanogels. The MIC values were similar with the ones calculated for loaded nanogel in the case of E. coli, but not for S. aureus. We highlight that the for the new experiment, the amoxicillin concentration matched the concentration of the drug loaded in BSA/MAC nanogels (0.166% w/v). The new data was not included in the manuscript.
Figure 1. Antibacterial activity of amoxicillin 0.166% against S. aureus (left) and E. coli (right).
|
Strain |
Amoxicillin 0.166% - inhibition zone (mm) |
MIC(mg/ml) |
MBC(mg/ml) |
|
S. aureus |
36.771±0.228 |
0.415 |
0.83 |
|
E. coli |
13.871±3.903 |
0.83 |
- |
A similar system was obtained by our group and already published in Biomacromolecules journal, were for the antimicrobial activity assessment of the new system, the unloaded nanogels were used as positive control.
